inorganic chemistry

hydrogen peroxide, manganese carbonyl, CO releasing moiety, redox reaction

**Authors for correspondence:**
Jacob A. Barrett
e-mail: jacob.alexander.barrett@gmail.com
Peter C. Ford
e-mail: pcford@ucsb.edu

[†]Present address: Department of Chemistry and Biochemistry, University of California, San Diego, San Diego, CA 92093 USA.

This article has been edited by the Royal Society of Chemistry, including the commissioning, peer review process and editorial aspects up to the point of acceptance.

# Redox-mediated carbon monoxide release from a manganese carbonyl—implications for physiological CO delivery by CO releasing moieties

Jacob A. Barrett[1,†], Zhi Li[1], John V. Garcia[1], Emily Wein[1], Dongyun Zheng[1], Camden Hunt[1], Loc Ngo[1], Lior Sepunaru[1], Alexei V. Iretskii[2] and Peter C. Ford[1]

[1]Department of Chemistry and Biochemistry, University of California, Santa Barbara, CA 93106, USA
[2]Department of Chemistry and Environmental Sciences, Lake Superior State University, Sault Sainte Marie, MI 49783, USA

JAB, 0000-0002-7522-5699; DZ, 0000-0003-0124-670X;
LN, 0000-0001-9476-5335; LS, 0000-0002-4716-5035;
AVI, 0000-0002-0226-5788

The dynamics of hydrogen peroxide reactions with metal carbonyls have received little attention. Given reports that therapeutic levels of carbon monoxide are released in hypoxic tumour cells upon manganese carbonyls reactions with endogenous $H_2O_2$, it is critical to assess the underlying CO release mechanism(s). In this context, a quantitative mechanistic investigation of the $H_2O_2$ oxidation of the water-soluble model complex *fac*-[Mn(CO)$_3$(Br)(bpCO$_2$)]$^{2-}$, (**A**, bpCO$_2^{2-}$ = 2,2′-bipyridine-4,4′-dicarboxylate dianion) was undertaken under physiologically relevant conditions. Characterizing such pathways is essential to evaluating the viability of redox-mediated CO release as an anti-cancer strategy. The present experimental studies demonstrate that approximately 2.5 equivalents of CO are released upon $H_2O_2$ oxidation of **A** via pH-dependent kinetics that are first-order both in [**A**] and in [$H_2O_2$]. Density functional calculations were used to evaluate the key intermediates in the proposed reaction mechanisms. These pathways are discussed in terms of their relevance to physiological CO delivery by carbon monoxide releasing moieties.

# 1. Introduction

The present investigation was triggered by reports that carbon monoxide can be released in tumour cells via the reaction of manganese carbonyl complexes with endogenous hydrogen peroxide [1–4]. Carbon monoxide delivery to biomedical targets using chemically activated CO releasing moieties (CORMs) [5–10] has drawn considerable therapeutic interest including applications in cancer treatment. Given that hypoxic tumours have significantly higher $H_2O_2$ concentrations than normal tissues [11–16], and that $H_2O_2$ and other reactive oxygen species may accumulate in certain cellular locations, CORM reactivity with $H_2O_2$ has the potential of enhancing targeted CO delivery. Similar activation of metal carbonyl-based CORMs has been noted with other tissues subject to oxidative stress [17,18]. Thus, this type of CORM activation has likely application in treating other inflammation sites, such as those infected by antibiotic-resistant bacteria [18–24].

Carbon monoxide is produced endogenously in mammals as a product of the catabolism of haems by the enzyme haem oxygenase-1. Ferrous ion and biliverdin are also formed, and all three can be cytoprotective (antiapoptotic, antioxidant, anti-inflammatory) [25,26]. Carbon monoxide releasing molecules have been previously shown to activate nuclear factor-2–erythroid related factor 2 (Nrf2) which regulates antioxidant defense [25–28]. Rotko et al. [29] found that large-conductance calcium-activated channels in inner mitochondrial membrane $BK_{ca}$ were inhibited by metal CORMs and haem. Overall, unravelling the reaction mechanisms of metal CORMs, and the consequent CO release in cells under oxidative stress, is essential to understanding and controlling the observed physiological effects.

Although there have been numerous studies of $H_2O_2$ reactions with dissolved metal ions and complexes [30–35], relatively little is known about the corresponding reactivity with metal carbonyls. $H_2O_2$ triggered CO release from the carbonyl complexes of manganese has been shown by He and co-workers [1,2]; however, the chemical mechanism(s) are undefined. Oxidation of the low-valent metal centres characteristic of carbonyl complexes should lead to CO release owing to reduced metal-to-CO $\pi$-backbonding, but many questions arise. For example: is the first step a one-electron, outer sphere oxidation also generating hydroxyl radicals? Or is it governed by an oxygen atom transfer (OAT) pathway? In addition, given that $H_2O_2$ can serve both as an oxidant and a reductant, does this initial step involve oxidation or reduction of the metal centre? Alternatively, is the first step a substitution reaction, where a peroxide species replaces a ligand on the metal centre? Since $H_2O_2$ is partially dissociated at physiological pH [36], is the active species $H_2O_2$ or the hydroperoxyl anion $HOO^-$? What are the products besides CO? What plausible reaction intermediates are formed? Also, are the reaction rates sufficient to explain the CO release under physiological conditions of oxidative stress? These questions are particularly relevant to therapeutic strategies based on such a reaction. Intratumoural selectivity using the variability of endogenous hydrogen peroxide concentrations may enable selective chemotherapy of cancer cells by CORMs [1–3].

The present work was initiated with the goal of addressing such mechanistic questions by interrogating the $H_2O_2$ oxidation of the Mn(I) model compound [37] fac-Mn(CO)$_3$(Br)(bpCO$_2$H) (bpCO$_2$H = 2,2′-bipyridine-4,4′-dicarboxylic acid) in near-neutral aqueous medium, where it is soluble as the [fac-Mn(CO)$_3$(Br)(bpCO$_2$)]$^{2-}$ dianion (**A**, equation (1.1)).

$$\tag{1.1}$$

Several similar Mn(I) carbonyls with the general motif Mn(CO)$_3$(X)L have been investigated as potential CORMs [38–40] and photoCORMs [41–48], and it is notable that **A** is also photoactive toward CO release [49]. In the present study, the reaction dynamics and products from the interaction of $H_2O_2$ with this model complex are elucidated, and possible reaction mechanisms are delineated in the context of density functional (DFT) computations.

# 2. Material and methods

## 2.1. Materials

Bromopentacarbonylmanganese(I) (98%) was purchased from Strem Chemicals. Coumarin-3-carboxylic acid (3-CCA) was purchased from Sigma-Aldrich. 4,4′-Dicarboxy-2,2′-bipyridine (98%), phosphoric

acid (85%), dimethyl sulfoxide, dibasic sodium phosphate and monobasic potassium phosphate were purchased from Fischer Scientific. Toluene, methanol, diethyl ether, dichloromethane, acetonitrile and tetrahydrofuran were purchased from Fischer Scientific, dried with 3 Å molecular sieves, and stored in an Argon glovebox. $H_2O_2$ (30%, certified ACS) was purchased from Fischer Scientific and titrated using the iodometric method to determine a stock solution concentration of 10.4 M. In all cases, diluted solutions were made using 18 MΩ cm resistivity deionized water. All deuterated solvents were purchased from Cambridge Isotope Laboratories. Carbon monoxide, carbon dioxide, nitrogen, argon and helium gas cylinders were purchased from Praxair.

The manganese(I) complex *fac*-[Mn(CO)$_3$(Br)(bpCO$_2$H)] was prepared as described by Cowan and co-workers [37,50]. A methanol solution of 2,2′-bipyridine-4,4′-dicarboxylic acid was added to a toluene solution of Mn(CO)$_5$Br and refluxed 2 h under an argon atmosphere, during which the colour of the solution changes from orange to red. Cooling and storing at 10°C overnight leads to the formation of a red solid precipitate that was collected by filtration then washed with diethyl ether and dichloromethane. The bright red powder was analysed by $^1$H NMR, IR and UV–Vis spectroscopy to ensure purity. $^1$H NMR (500 MHz, DMSO-d$_6$): δ (ppm) 8.10 (d, 2H), 9.04 (s, 2H), 9.40 (d, 2H). $^1$H-NMR (500 MHz, D$_2$O): δ (ppm) 7.97 (d, 2H), 8.70 (s, 2H), 9.36 (d, 2H). FTIR (95 : 5 DMSO/H$_2$O) 2022, 1936, 1920 cm$^{-1}$. The NMR, IR and quantitative UV–Vis spectra observed were consistent with the reported values [37,50].

## 2.2. Instrumentation and analytical procedures

Optical spectra and most kinetics were recorded on a Shimadzu UV-2401PC spectrophotometer. Actions were taken to limit any exposure of the cell to light. For example, most of the overhead room lights were off and the two remaining lights are equipped with amber filters. In addition, the spectrometer was covered with a black cloth to shield from the light from the monitor. Liquid phase infrared spectra were recorded with a Nicolet iS50 FTIR spectrometer in solution cells with CaF$_2$ windows. Perpendicular-mode X-band EPR spectra were recorded for samples in quartz tubes using a Bruker EMX EPR spectrometer equipped with an Oxford ESR 900 liquid helium cryostat. Aqueous samples were measured using a capillary insert to minimize the cross-sectional area of the sample. Low-temperature samples were frozen in a dry ice/ethanol bath prior to placement in the sample holder. These EPR samples were shielded from light using a black cloth up until the point of measurement.

Carbon monoxide and carbon dioxide releases from the reaction of **A** with $H_2O_2$ were quantified using an Agilent 6890N (G1530N) gas chromatograph equipped with a thermal conductivity detector (GC-TCD) and a Sigma-Aldrich Carboxen-1006 capillary column (30 m × 0.32 mm with average thickness 15 μm) [49]. Samples were obtained using a gas-tight syringe to sample the headspace of a screw-top septum quartz cuvette and injected into the inlet in splitless mode. The inlet pressure and temperature were 0.5 bar and 225°C. The carrier gas, helium, was set at a constant flow rate of 1 ml min$^{-1}$. The column temperature was held at 35°C for 20 min, then ramped to 250°C at 10°C min$^{-1}$ where it was held for 25 min. The thermal conductivity detector was set at 230°C with a 12 ml min$^{-1}$ reference flow rate and 7 ml min$^{-1}$ He make-up flow rate. Gas-phase products (CO and $CO_2$) were quantified by comparison with calibration curves generated using a Schlenk flask filled with a known pressure (measured by a mercury manometer) of the gas obtained from a cylinder. Injection volumes were between 10 and 200 μl (10–50 μl for the low CO concentrations on the calibration curve and 200 μl for reaction headspace analysis).

Luminescence spectra were obtained using a Photon Technology International fluorimeter with an 814 PMT detection system at 1 nm resolution in a darkroom. The excitation and emission slit width were both set to 5 nm. 3-CCA was used as a probe for hydroxyl radical formation where 3-CCA is converted to 7-hydroxy-coumarin-3-carboxylic acid (7-OH-3-CCA) with a 4.7% conversion factor [51]. The emission spectrum of 7-OH-3-CCA was recorded from 420 to 520 nm using a 395 nm excitation wavelength. A 400 μl aliquot of a 2 mM 3-CCA solution, a 60 μl aliquot of a solution of **A** (1.9 mM), and 1.78 ml aliquot of aqueous phosphate buffer solution (118 mM) were added to a quartz cuvette. Then, a 20 μl aliquot of $H_2O_2$ stock solution was added to give a final concentration of 32 mM $H_2O_2$ in the cuvette. The solutions were stirred at 24°C for 3 h before recording the fluorescence spectrum. Control reactions were performed without Mn and/or without $H_2O_2$ as well as with CuCl$_2$, which is known to catalyse the Fenton reaction [52]. Absorbance changes were observed on the Shimadzu spectrophotometer with lower 3-CCA concentration (100 μl aliquots).

Electrochemical measurements were made using a Metrohm PGSTAT 128N potentiostat. The working electrode was glassy carbon (R = 3 mm), the counter electrode was a graphite rod and the reference was a saturated calomel electrode (SCE). All ICP-AES measurements were obtained using a ThermoiCAP 6300 ICP equipped with a 6000 K Ar plasma. Powder XRD patterns were collected under ambient temperature

using a Panalytical Empyrean diffractometer, with Cu K$\alpha$ radiation ($\lambda = 1.5405980$ Å) in a stainless-steel sample holder, scanned from $5°$ to $75°$ ($2\theta$).

## 2.3. Kinetics studies

The reaction rates were determined from temporal absorbance changes using the Shimadzu UV-2401PC spectrophotometer with the samples protected from exposure to light as noted above. The method of initial rates was used for the reaction of **A** with $H_2O_2$ where the rates were determined by linear regression of the first 5% decrease in absorbance. A typical reaction solution at different [**A**] was prepared by adding an aliquot (50–200 µl) of a stock solution of **A** (1.9 mM) to an aliquot (2.10–2.25 ml) of aqueous phosphate buffer (118 mM) in a quartz cuvette. This was then allowed to equilibrate in the thermostated spectrophotometer cell holder for at least 15 min. Then, a 20 µl aliquot of a hydrogen peroxide stock solution was added to give a final [$H_2O_2$] of 8–32 mM and the decrease in the metal-to-ligand charge transfer (MLCT) band of **A** was monitored. Most kinetics runs were carried out in aerobic media, although, in some cases, the cuvette was entrained with $N_2$ before adding the peroxide. There were no significant differences in the initial rates kinetics between aerobic and $N_2$ entrained solutions.

## 2.4. Density functional theory methods

Geometry and frequency calculations were performed with Gaussian 16 software package. The ground state structures were optimized at spin unrestricted B3LYP/DGDZVP level of theory without symmetry constraints, then the time-dependent DFT was used to obtain relative energy changes ($\Delta E$, kcal mol$^{-1}$). All calculations in the ground and excited states were completed in the presence of a solvent (water) using integral equation formalism of the polarizable continuum model (IEFPCM). All energies are reported with the thermal correction to 298 K.

# 3. Results

## 3.1. Reaction products

A phosphate-buffered solution of **A** is yellow. Under these conditions (pH 7.4, buffer concentration 118 mM), the carboxylate groups of **A** are fully deprotonated given the previously determined p$K_a$ values of 2.25 and 2.6 [37]. The optical spectrum (figure 1, electronic supplementary material, figure S1) displays a broad visible/near UV absorption band with $\lambda_{max}$ 408 nm ($\varepsilon = 3.3 \times 10^3$ M$^{-1}$ cm$^{-1}$), which, by analogy to other Mn(I) complexes of polypyridine ligands, can be assigned to a MLCT transition. The strong UV bands at 300 and 220 nm are attributed to $\pi$–$\pi^*$ transitions characteristic of the conjugated rings of the bpCO$_2^{2-}$ ligand (bpCO$_2^{2-}$ = 2,2′-bipyridine-4,4′-dicarboxylate dianion). This spectrum is stable for hours in buffered, aerobic aqueous solution in the dark. Figure 1 also displays temporal spectral changes upon adding excess $H_2O_2$ that are distinctly indicated by decreasing MLCT absorbance. Qualitatively, the result is bleaching of the yellow colour and some accompanying bubble formation. The final spectrum was close to that of the free ligand bpCO$_2$H dissolved in a pH 7.4 buffered solution ($\lambda_{max}$ 294 nm, $\varepsilon = 12.3 \times 10^3$ M$^{-1}$ cm$^{-1}$, electronic supplementary material, figure S1) at the original concentration of **A**.

The headspace above the reaction solution was sampled concurrently with the temporal $H_2O_2$-induced spectral changes shown in figure 1. Gas chromatographic (GC) analysis showed that after 1 h, the CO release corresponded to 1.1 moles per mole of **A** consumed (based on decreased absorbance at 408 nm, Abs$_{408}$). After 3 h, this was 1.7 moles of CO per mole of **A** reacted. After 5 h, Abs$_{408}$ was fully depleted and GC analysis showed about 2.3 moles of CO released per mole of **A** reacted (electronic supplementary material, figure S2). No further change in Abs$_{408}$ or CO release was observed after another 18 h. The resulting solution was then acidified with $H_3PO_4$, and GC analysis of headspace indicated the release of approximately 0.6 moles $CO_2$ per mole of **A** reacted (electronic supplementary material, figure S2).

Performing the reaction with more concentrated **A** (2 mM) in pH 7.4 phosphate buffer (118 mM) led to the formation of a white precipitate. ICP analysis of this product gave a Mn to P molar ratio of 1.45 consistent with that of manganese phosphate Mn$_3$(PO$_4$)$_2$. The XRD pattern for the isolated powder

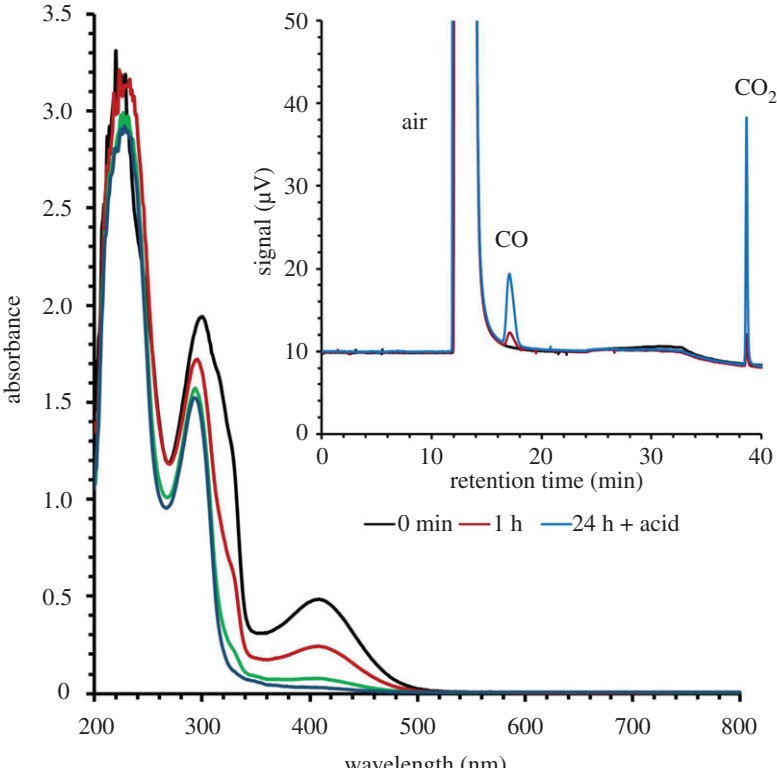

**Figure 1.** The reaction of **A** with $H_2O_2$. Black: the optical spectrum of a phosphate buffer (pH = 7.4, 118 mM) solution of **A** ($1.5 \times 10^{-4}$ M). Red: spectrum 1 h after addition of $H_2O_2$ (32 mM). Green: after 3 h. Blue: after 5 h. $T = 36.8°$ C. Inset: gas chromatograms of 200 µl samples of the headspace taken after 1 h (red) and after 24 h and acidification to release $CO_2$ (blue) (see electronic supplementary material, figure S2).

gave the Bragg reflections consistent with those reported for $Mn_3(PO_4)_2 \cdot 3H_2O$ (electronic supplementary material, figure S3) [53].

The $^1$H NMR spectrum **A** in pH 7.4 phosphate-buffered $D_2O$ displayed sharp resonances for the aromatic protons at 7.97, 8.70 and 9.36 ppm of the coordinated $bpCO_2^{2-}$ ligand (electronic supplementary material, figure S4). Adding $H_2O_2$ led to the rapid intensity decrease and broadening of all NMR peaks, presumably owing to the formation of labile, paramagnetic manganese species. No additional features were observed between 80 and −80 ppm.

*In situ* X-band electron paramagnetic resonance (EPR) spectra recorded for an aqueous solution of **A** (150 µM) showed no signal, further confirming a diamagnetic complex. After reaction with added $H_2O_2$ (32 mM) a broad signal was observed for the room temperature solution, which did not display any further details even after careful freezing to 100 K (electronic supplementary material, figure S5). When the reaction was carried out in acetonitrile, a six-line spectrum was obtained (electronic supplementary material, figure S6) consistent with the expected spectrum for a $d^5$ high spin Mn(II) ion. Thus, we conclude that the reaction of **A** with $H_2O_2$ leads to the formation of Mn(II) and the free ligand $bpCO_2^{2-}$, and that after reaction completion, approximately 2.3 moles of CO plus approximately 0.6 moles of $CO_2$ are released per mole of **A** consumed (equation (3.1)).

$$\mathbf{A} + H_2O_2(xs) \rightarrow Mn^{2+} + bpCO_2^{2-} + \sim 2.3\,CO + \sim 0.6\,CO_2. \qquad (3.1)$$

## 3.2. Electrochemical measurements

The reaction was further probed by measuring the formal reduction potentials of the compound voltammetrically and by detecting possible products when the complex was exposed to $H_2O_2$. Cyclic voltammetry of **A** in pH 7.4 phosphate-buffered solution at 37°C shows an irreversible oxidation at approximately 0.76 V and an irreversible reduction at about −1.11 V versus a SCE (electronic supplementary material, figure S7). The latter value is consistent with an earlier measurement in 95%/5% acetonitrile/water (−1.07 V) [37]. Some bubble formation was observed upon the reaction of

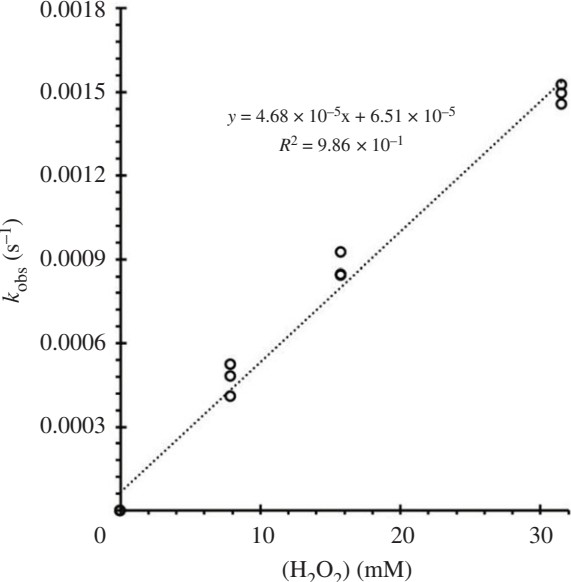

**Figure 2.** Plot of $k_{obs}$ ($s^{-1}$) versus $[H_2O_2]_{tot}$ in aerobic phosphate buffer (118 mM) at pH 7.4 and 36.8°C. $k_{obs}$ values ($k_{obs} = R_i/[\mathbf{A}]_i$) were derived using the initial rate method by following the disappearance of the MLCT absorbance at 408 nm. Various initial concentrations $[\mathbf{A}]_i$ were between 34 and 147 μM. Each point represents an independent experiment. The slope of this plot is $4.7 \times 10^{-2}$ $M^{-1}$ $s^{-1}$.

**A** with $H_2O_2$. While CO release might be partially responsible, another candidate would be $O_2$, given that Mn(II) salts are reported to catalyse $H_2O_2$ disproportion [53,54]. In this context, $O_2$ formation was detected by reductive scans from 0 to −0.85 V (electronic supplementary material, figure S8). The irreversible peak at approximately −0.55 V corresponds to the two-electron reduction of $O_2$ to $H_2O_2$ on a glassy carbon electrode [55,56], reflecting that $O_2$ is generated as a by-product.

## 3.3. Reaction dynamics

As shown in figure 1 and electronic supplementary material, figure S9, the addition of $H_2O_2$ to a buffered solution of **A** leads to progressive decreases in the visible range MLCT absorbance to give a spectrum largely attributed to that of the free ligand. No isosbestic points were observed, and there were no spectral indications of any intermediates. Similarly, when the reaction was followed using FTIR spectroscopy (the solvent in that case being 95 : 5 dimethyl sulfoxide: phosphate buffer), the $\nu_{CO}$ bands at 2022, 1936 and 1920 $cm^{-1}$ characteristic of **A** decreased slowly with no other metal carbonyl species being evident (electronic supplementary material, figure S10).

In this context, the reaction rates were evaluated by following the disappearance of the MLCT band at the 408 nm $\lambda_{max}$ characteristic of the Mn(I) complex. The method of initial rates was employed to evaluate the reaction kinetics (see Material and methods section), owing to the bubble formation in the latter stages of the reaction. As shown in electronic supplementary material, table S1, the initial rate $R_i$ is directly proportional to $[\mathbf{A}]_i$, the initial concentration of **A**, under (pseudo) first-order conditions (large excess of $H_2O_2$) (equation (3.2)). Although there was some scatter in the data owing to the greater uncertainty of the initial rate method, this behaviour proved true for each set of conditions studied (electronic supplementary material, table S1 and figure S11).

$$R_i = -\left(\frac{d[\mathbf{A}]}{dt}\right)_{initial} = k_{obs}[\mathbf{A}]_i. \tag{3.2}$$

Figure 2 displays the plot of the apparent rate constant $k_{obs}$ (obtained from dividing the initial rate by $[\mathbf{A}]_i$) versus $[H_2O_2]_{tot}$, where $[H_2O_2]_{tot}$ represents the total peroxide initially present at pH 7.4. Thus, for these conditions where $[H_2O_2]_{tot} \gg [\mathbf{A}]_i$, we can rewrite equation (3.2) as

$$R_i = k'[H_2O_2]_{tot}[\mathbf{A}]_i, \tag{3.3}$$

where $k_{obs} = k'$ $[H_2O_2]_{tot}$ and $k'$ is the apparent second-order rate constant for the rate-limiting step.

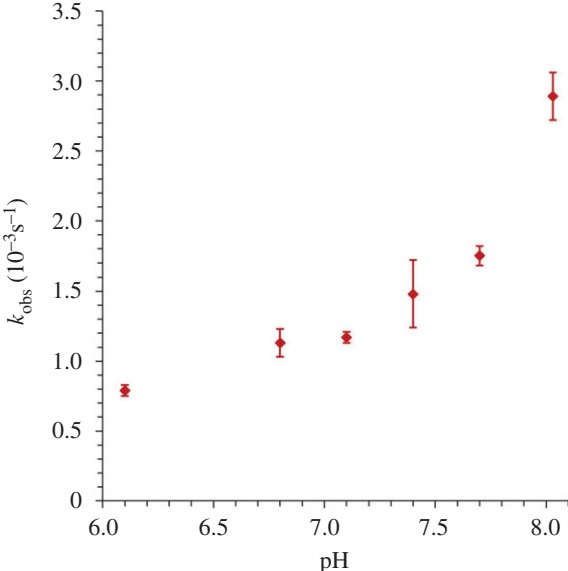

**Figure 3.** Plot of $k_{obs}$ versus pH for the reaction of **A** (at var. concentration) with $H_2O_2$ ([$H_2O_2$]$_{total}$ = 31.5 mM) as measured for the disappearance of **A** at 408 nm at 36.8°C. Each point is an average of at least three to five experiments. $k_{obs} = R_i/[\mathbf{A}]_i$.

Electronic supplementary material, figure S11 displays similar linear plots of $k_{obs}$ versus [$H_2O_2$]$_{tot}$ for temperatures over the range 292.7–315.2 K at pH 7.4. The slopes of these plots gave values of $k'$ as a function of temperature. An Eyring plot of the latter data (electronic supplementary material, figure S12) was used to calculate the apparent activation enthalpy ($\Delta H^{\ddagger} = 22 \pm 3$ kcal mol$^{-1}$) and activation entropy ($\Delta S^{\ddagger} = 7 \pm 3$ cal mol$^{-1}$ K$^{-1}$) giving the effective temperature dependence at this pH and under these conditions. For studies in acetate buffer at pH 5, the apparent activation enthalpy proved to be somewhat lower (approx. 16 kcal mol$^{-1}$) as calculated from the data in electronic supplementary material, table S1D.

Given that $H_2O_2$ is a weak acid (equation 3.4) with a $pK_a$ of 11.62 [36], a key question would be whether such acid–base behaviour will be reflected in the kinetics of the reaction with **A** since the reactivity of $H_2O_2$ and its conjugate base (HOO$^-$) should differ. Figure 3 displays the pH dependence of the $R_i$ as a function of pH over the phosphate buffer limited range 6.1–8.1. These data clearly indicate that the rates are only modestly sensitive to pH at the lower values studied, but the rate accelerates with increasing pH, suggesting that the hydroperoxyl anion HOO$^-$ is markedly more reactive with **A** than is $H_2O_2$ itself.

$$H_2O_2 \rightleftharpoons HOO^- + H^+ \qquad K_a = 2.4 \times 10^{-12} \text{ M,} \tag{3.4}$$

Thus, we can rewrite the $k_{obs}$ of equation (3.2) to give

$$k_{obs} = k_1[H_2O_2] + k_2[HOO^-], \tag{3.5}$$

and this equation can be rewritten in terms of [$H_2O_2$]$_{tot}$ as follows:

$$k_{obs} = \left(\frac{k_1[H^+] + k_2 K_a}{[H^+] + K_a}\right)[H_2O_2]_{tot}, \tag{3.6}$$

which simplifies to

$$k_{obs} = \left(k_1 + \frac{k_2 K_a}{[H^+]}\right)[H_2O_2]_{tot}, \tag{3.7}$$

when [H$^+$] $\gg K_a$, as in the present case. Accordingly, a plot of $k_{obs}$ versus [H$^+$]$^{-1}$ should be linear with an intercept equal to $k_1$[$H_2O_2$]$_{tot}$ and a slope equal to $k_2 K_a$[$H_2O_2$]$_{tot}$. This behaviour is demonstrated in electronic supplementary material, figure S13 for data obtained at 36.8°C in phosphate buffer, and from this dataset, the values of $247 \pm 20$ M$^{-1}$ s$^{-1}$ for $k_2$ and $2.8 \pm 0.2 \times 10^{-2}$ M$^{-1}$ s$^{-1}$ for $k_1$ were calculated. The latter value, determined by extrapolation, is similar to that determined at pH 5.0 in a different buffer ($2.0 \pm 0.2 \times 10^{-2}$ M$^{-1}$ s$^{-1}$ in acetate/acetic acid, electronic supplementary material,

table S1D). However, the key result is that both hydrogen peroxide and its conjugate base $HOO^-$ are reactive toward this manganese(I) carbonyl with the latter being approximately $10^4$ times more reactive than $H_2O_2$. While both forms contribute to the overall reaction rate at physiologically relevant near-neutral pH values, the greater contribution at pH 7.4 or lower would be from $H_2O_2$. At higher pH the reaction with $HOO^-$ would predominate.

## 3.4. Possible mechanisms

One can infer from the kinetics data that the rate-determining step(s) is (are) second-order: first-order in $[A]_i$ and first-order in $[H_2O_2]_{tot}$, although the latter reflects the activity of both $H_2O_2$ and its conjugate base $HOO^-$. At 36.8°C in pH 7.4 buffered solution, the apparent second-order rate constant $k'$ was determined to be $(5.3 \pm 1.0) \times 10^{-2} \, M^{-1} \, s^{-1}$ (electronic supplementary material, table S1A). Three prospective mechanistic schemes for the reaction in near-neutral aqueous media are presented in electronic supplementary material, schemes S1A–C. These can be categorized in terms of the proposed first steps: outer sphere electron transfer, ligand substitution, or attack on a coordinated CO to give a hydroperoxyl–carbonyl complex. We will consider these proposals individually in the context of the kinetics experiments and DFT calculations.

Outer sphere electron transfer from A to $H_2O_2$ (or $HOO^-$) (scheme 1), $R = CO_2^-$; $X^- = Br^-$ or triflate would give the analogous manganese(II) complex first formed as the low spin $3d^5$ doublet $^2B$. Intersystem crossing to the high spin $d^5$ sextet state $^6B$ would lead to rapid carbon monoxide release, as illustrated in electronic supplementary material, scheme S1A, owing to markedly decreased metal to CO $\pi$-backbonding. Consistent with this view was the finding that DFT calculations for $^6B$ at B3LYP/DGDZVP level of theory did not converge to a stable species.

The feasibility of the outer sphere electron transfer can be assessed from the electrochemical properties of A and $H_2O_2$. As reported above, the irreversible anodic wave for the oxidation of A was measured as approximately 0.76 V (versus SCE) in pH 7.4 buffer solution (i.e. approx. 1.0 V versus NHE) for the half-cell $A \rightarrow B + e^-$. $H_2O_2$ is a strong two-electron oxidant, with a standard reduction potential ($E_o'$) of 1.32 V at pH 7.0 [55,56], but it is a weak one-electron oxidant (0.39 V). Thus, $\Delta E$ for scheme 1 would be unfavourable by nearly 0.7 V. Similarly, our DFT computations (with solvation corrections using the polarizable continuum model) found the reaction in scheme 1 unfavourable by 66 kcal mol$^{-1}$ (electronic supplementary material, table S2A). Furthermore, if one presumes that $H_2O_2$ and $HOO^-$ react by similar mechanisms, the much greater reactivity of the latter with A clearly excludes the outer sphere electron transfer mechanism. Thus, it appears very unlikely that this would be the first step in a sequence of reactions leading to the eventual products.

The other predicted product of scheme 1 is the hydroxyl radical ($\bullet OH$). As electronic supplementary material, scheme S1A illustrates, it is likely that in the absence of other trapping agents the $\bullet OH$ would either oxidize A or react with $H_2O_2$ itself to give the hydroperoxyl radical $\bullet OOH$. [57] Given the much higher concentration of $H_2O_2$, the latter pathway would probably dominate. As hypothesized in electronic supplementary material, scheme S1A, this would lead to oxidation of another equivalent of A, concomitant with releasing two more COs plus a $CO_2$ via intermediates C and D. The predicted 5/1 ratio of $CO/CO_2$ released is roughly consistent with the stoichiometry observed. DFT computations confirm that these subsequent steps would be favourable (electronic supplementary material, table S2A).

Formation of $\bullet OH$ can be probed by trapping with coumarin-3-carboxylic acid (3CCA) to give the fluorescent compound 7-hydroxy-coumarin-3-carboxylic acid [51,58]. The sensitivity of this probe was tested using the Fenton-like reaction of $H_2O_2$ (32 mM) catalysed by $CuCl_2$ (45 µM) in the presence of added 3CCA (86 µM in 25°C, pH 7.4 buffer) [52]. The photoluminescence (PL) spectrum of the resulting solution showed PL at $\lambda_{max}$ approximately 450 nm, but none was observed with $CuCl_2$ alone in the absence of $H_2O_2$ (electronic supplementary material, figure S15). When the same test was

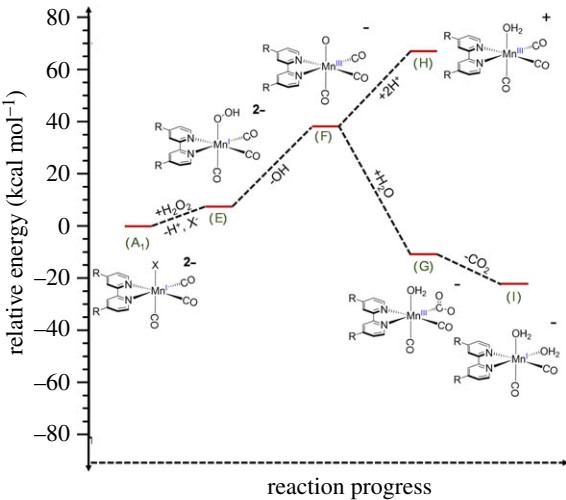

**Figure 4.** Graphical representation of proposed substitution pathway for the reaction of **A** with hydrogen peroxide. The relative energies are based on the DFT calculations shown in electronic supplementary material, table S2.

carried out with $H_2O_2$ (32 mM) and **A** (45 µM), a very weak PL band was observed, only marginally stronger than when the test was done with $H_2O_2$ alone (electronic supplementary material, figure S15). Given the unfavourable energetics of the reaction in scheme 1, this small difference may reflect minimal •OH generation during $H_2O_2$ disproportionation catalysed by the reaction products (see above). Notably, this result is consistent with that by Poole and co-workers [22] who observed •OH upon UV irradiation of the Mn(I) carbonyl $[Mn(CO)_3(tpa-\kappa^3N)]Br$ (tpa = tris(2-pyridylmethyl)amine in the presence of $H_2O_2$ but saw none from thermochemical processes.

A seemingly more likely pathway would be an inner sphere process involving ligand substitution as the first step (figure 4 **A–E**, also electronic supplementary material, scheme S1B). The redox step would occur subsequently via $OH^-$ dissociation from $Mn(CO)_3(OOH)(bpCO_2)]^{2-}$ (**E**) to give the Mn(III) oxo complex $Mn(CO)_3(O)(bpCO_2)]^-$ (**F**). This would be followed by competing sequential steps such as proposed in electronic supplementary material, scheme S1B to give Mn(II), free CO and $CO_2$ as well as free $bpCO_2^{2-}$. However, the net stoichiometry of these steps with regard to the $CO/CO_2$ ratio is less clear-cut, ranging from 2/1 to 5/1 depending on how the proposed intermediates partition along the various reaction channels suggested by this scheme.

A DFT evaluation of the first step shows this to have a modestly unfavourable $\Delta E$ (7 kcal mole$^{-1}$). However, the energetics should be pH dependent, and indeed the reaction of **A** with $HOO^-$ was calculated to be exoergic ($\Delta E = -26$ kcal mol$^{-1}$) (electronic supplementary material, table S2A). Such calculations help in evaluating whether the respective intermediates are energetically accessible. Although DFT calculations suggest that reaction of **A** with either $H_2O_2$ or hydroperoxyl anion $HOO^-$ to give **E** is a reasonable first step, the next step in figure 4 and electronic supplementary material, scheme S1B, where the manganese centre is formally oxidized, is less favourable (+31 kcal mol$^{-1}$). Again, the calculated energetics of this step are likely to be pH dependent, and given the apparent $\Delta H^{\ddagger}$ of $22 \pm 3$ kcal mol$^{-1}$, this pathway should be reasonable, although this model suggests that the second step, **E–F**, would be the rate-limiting step.

The substitution lability of **A** was probed experimentally by evaluating the kinetics of the reaction with pyridine (py) in pH 7.4 phosphate buffer solution at 25°C. The spectroscopic result showed a modest shift of the MLCT band to $\lambda_{max}$ 400 nm and a corresponding increase in the absorbance at 299 nm (electronic supplementary material, figure S16) over a period of several hours. We attribute these changes to the formation of *fac*-$[Mn(CO)_3(py)(bpCO_2)]^-$ (equation (3.8)). The observed rate constants $k_{obs}$(subs) monitored by absorbance increase at 299 nm were nearly linearly dependent on the pyridine concentration over the range 32–200 mM (electronic supplementary material, table S3). Dividing $k_{obs}$(subs) by [py] gave a second-order rate constant of approximately $2 \times 10^{-2}$ M$^{-1}$ s$^{-1}$ (electronic supplementary material, table S3) nearly identical to the second-order rate constant $1.7 \pm 0.4 \times 10^{-2}$ M$^{-1}$ s$^{-1}$ measured under comparable conditions (25.5°C, pH 7.4) for the reaction of **A** with $H_2O_2$ (electronic supplementary material, table S1C). Thus, the substitution-driven mechanism initiated by **A** with $H_2O_2$ (figure 4) appears feasible.

$$\mathbf{A} + py \xrightarrow{k_{obs}(subs)} Mn(CO)_3(py)(bpCO_2)]^- + Br^-. \tag{3.8}$$

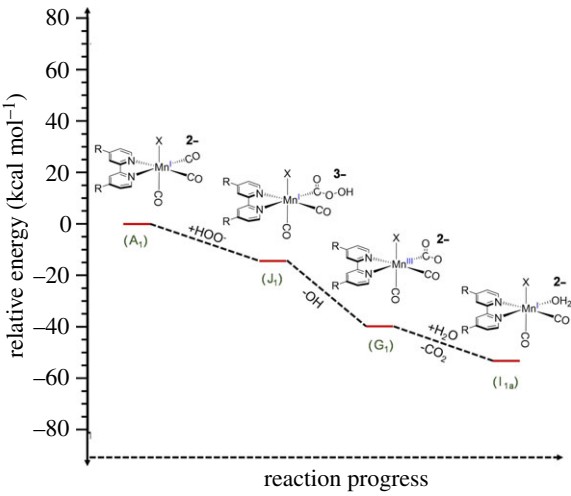

**Figure 5.** Graphical representation of the proposed nucleophilic activation of **A** by HOO⁻ reaction with a CO on **A**. The relative energies are based on the DFT calculations shown in electronic supplementary material.

Further evidence favouring this mechanism derives from the following experiment. The spectrum of the solution resulting from a reaction of py with **A** proved stable for at least 15 h. The addition of $H_2O_2$ led to temporal spectral changes (electronic supplementary material, figure S17) indicating oxidation of the manganese complex in analogy to the reaction of **A** with $H_2O_2$ (equation (3.1)). Like the latter reaction, the rate of decay was dependent on $[H_2O_2]$, but the second-order rate constant ($k_{obs}$(ox)/$[H_2O_2]$) was an order of magnitude lower (approx. $0.2 \times 10^{-2}\,M^{-1}\,s^{-1}$) compared with complex **A**. Furthermore, this value did not change appreciably over a six-fold range in added py (electronic supplementary material, table S3). Thus, blocking the labile coordination site strongly inhibits the reactivity with $H_2O_2$. These two lines of circumstantial evidence imply that ligand lability plays an important role in the oxidation of **A** by $H_2O_2$.

The third prospective mechanism outlined in electronic supplementary material, scheme S1C involves the formation of a hydroperoxycarbonyl intermediate (**J**) via attack of the peroxide at the coordinated carbonyl (figure 5). Such nucleophilic attack on a metal carbonyl has been demonstrated with methoxide [59], for example, and the reaction of coordinated CO with hydroxide is a key step in the metal carbonyl catalysed water gas shift reaction [60]. Furthermore, the hydroperoxyl anion is an unusually powerful nucleophile [61], so this pathway is particularly attractive for reaction with **A**. DFT computations find this step very favourable for HOO⁻ ($\Delta E = -13$ kcal mol⁻¹), as well as the subsequent steps leading from species **J** to $[BrMn^I\,(CO)_2L(H_2O)]^{2-}$ (**I**) + $CO_2$ + OH⁻ ($\Delta E = -41$ kcal mol⁻¹). However, the net result of this proposed reaction sequence is conversion of one CO to $CO_2$. Oxidation of **I** with $O_2$ or $H_2O_2$ such as those postulated in electronic supplementary material, scheme S1C may be required to labilize the remaining COs.

Nonetheless, the following observations argue for the possible role of this pathway in the overall scheme of reactions leading to oxidatively mediated CO labilization from **A**. The first is that the residual reactivity of $Mn(CO)_3(py)(bpCO_2)]^-$ with $H_2O_2$ proved independent of $[py]$, thus arguing against this reaction occurring via reversible dissociation of py. Another is that the reaction rate is about four times faster at pH 8.1 than at pH 7.4, again indicating the greater reactivity of HOO⁻. Given that this anion is a powerful nucleophile [61], its much greater reactivity toward **A** would not distinguish clearly between the redox pathway mediated by ligand substitution as in figure 4 (and electronic supplementary material, scheme S1B) and one initiated by nucleophilic attack on a coordinated CO as described in figure 5 (electronic supplementary material, scheme S1C). However, the former pathway appears to be excluded by the formation of the pyridine complex, leaving the latter as a likely player. In these contexts, we conclude that it is probable that both mechanisms play roles in the present system.

## 4. Discussion

The results described here clearly point to the importance of intracellular pH as a potent factor for controlling the location and rate of redox-mediated CO release from a metal-based CORM. Notably,

the localized pH varies; the lysosome is typically more acidic (pH 4.7) than the cytosol while the space between the mitochondria outer and inner membranes (mitochondrial matrix) is typically more basic (pH 8) [62]. Thus, if the peroxide concentrations were comparable in the lysosome and in the mitochondrial matrix, the reaction with the manganese carbonyl **A** would be approximately four times faster in the latter. Furthermore, we have shown pH-dependent reaction kinetics for complex **A**, which implies there are at least two functioning mechanisms. In the mitochondrial matrix, we would expect the formation of the proposed hydroperoxycarbonyl intermediate (figure 5) to dominate the reactivity, while in the lysosome, the ligand substitution pathway (figure 4) should have the higher contribution.

Certainly, pH alone will not determine the rate of CO delivery to specific targets given the complexity of such biological systems. One must consider the localization of $H_2O_2$ production and the diffusion of this reactant to other sites as well as the targeting of the relevant CORM. Szatrowski & Nathan [11], on the basis of studies with seven human tumour cell lines, estimated that $H_2O_2$ production may be up to 0.5 nmoles per $10^4$ cells $h^{-1}$. If one assumes cellular volumes of approximately 2 pl [63], this translates into approximately 25 mmoles $l^{-1} h^{-1}$. While there are numerous pathways that deplete $H_2O_2$, e.g. catalase [64], such estimates suggest that even the relatively slow reaction with the model complex **A** ($k' = \sim 5 \times 10^{-2} M^{-1} s^{-1}$ in pH 7.4 phosphate buffer at 37°C) has the potential to deliver physiologically relevant concentrations of CO, especially if this CORM were localized in the mitochondria.

Since the reaction does not generate appreciable quantities of hydroxyl radicals, the cytotoxic effects of Mn(I) carbonyls probably result from the interaction of the released CO with haem-containing proteins crucial to cellular respiration. For example, CO binding to cytochrome c oxidase would thereby affect the redox chain in mitochondria and increase oxidative stress. Thus, the biological effects of Mn(I) carbonyls would result from downstream effects of the released CO and perhaps the metal ion, rather than the Fenton-like generation of radicals.

Having insight into the potential mechanism(s) for hydrogen peroxide mediated CO release provides guidelines for the design of new CORMs that may function accordingly. The cellular pathways of a metal-based CORM such as **A** are undoubtedly dependent on where this species is localized in the cell. Thus, a very important design direction to explore is selective organelle uptake and localization. For example, a triphenylphosphonium moiety was used to increase the mitochondrial uptake of graphene nanocarriers of the small molecule bioregulator NO [65]. In addition, increasing the electrophilicity of a CORM based on the Mn(I) platform should enhance the reactivity toward $H_2O_2$. Ligand design has been extensively studied for tuning the spectra, solubility, imageability, reaction selectivity and cellular uptake. Given the greater reactivity at higher pH, incorporating a ligand with a pendent base that can assist $H_2O_2$ deprotonation might be a promising direction. The reactivity patterns established here point to other possibilities in terms of molecular engineering to explore the intracellular mechanisms and the effectiveness of redox-mediated CO release from CORMs. Several of these are under continuing investigation.

## 5. Conclusion

The model Mn(I) CORM *fac*-Mn(CO)$_3$(Br)(bpCO$_2^-$) (**A**) reacts with $H_2O_2$ in phosphate buffer to release CO and $CO_2$ in roughly a 5 : 1 mole ratio. At higher concentrations in phosphate buffer, the manganese product was identified as a Mn(II) phosphate salt. The Mn(II) product apparently catalyses $H_2O_2$ disproportionation, given that $O_2$ was observed via electrochemical methods. **A** is modestly substitution labile as shown by the reaction with pyridine. Furthermore, DFT calculations imply that substitution of HOO$^-$ for the bromide on **A** and nucleophilic attack on a coordinated carbonyl by HOO$^-$ are both reasonable initial steps in the sequence leading to the oxidation of the metal centre and CO release. Neither **A** nor the Mn(II) product appreciably catalyses Fenton chemistry to form •OH. The pH dependence of the kinetics indicates both $H_2O_2$ and HOO$^-$ to be reactive with **A**, but the latter is nearly four orders of magnitude more so.

Data accessibility. Additional information concerning this paper is available in the electronic supplementary material and in Dryad Digital Repository: https://doi.org/10.5061/dryad.mcvdnck1f. The electronic supplementary material information (16 figures, three tables and one set of schemes) includes electronic supplementary material, experimental results including product characterization by GC, XRD, NMR, IR and ESR studies; detailed kinetics results and DFT computational details as well as a scheme outlining potential mechanisms for product formation from the reaction of $H_2O_2$ with **A**.

The data are provided in the electronic supplementary material [66].

Authors' contributions. J.A.B. and P.C.F. conceived of the research project and served as the principal authors of the manuscript. Z.L. initially prepared compound **A** and participated in the first experimental studies; J.A.B., J.V.G., E.W. and L.N. conducted kinetics experiments, C.H. and J.A.B. conducted EPR studies; D.Z., L.S. and J.A.B. conducted electrochemical studies; A.V.I. carried out the DFT calculations. All authors discussed aspects of the results and had the opportunity to comment on the manuscript.

Competing interests. The authors declare no competing interests.

Funding. J.A.B. acknowledges a predoctoral fellowship from the US National Science Foundation Graduate Research Fellowship Program (award no. 1650114). This research was supported in part by a grant to P.C.F. from the Chemistry Division of the United States National Science Foundation (grant no. CHE-1565702). The authors acknowledge the MRL Shared Experimental Facilities which are supported by the NSF MRSEC Program under award no. DMR 1720256; a member of the NSF-funded Materials Research Facilities Network. DFT calculations were accomplished on computational facilities provided by a National Science Foundation grant (CNS-1725797) and administered by the UC Santa Barbara Center for Scientific Computing, which is supported by the California NanoSystems Institute and the UCSB Materials Research Science and Engineering Center (MRSEC; NSF DMR 1720256).

Acknowledgements. J.A.B. acknowledges Qin Jiang and Peng-Fi Shi for advice with synthetic preparation.

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
