## [Peer Review File · Royal Society Open Science]

Review History

RSOS-211022.R0 (Original submission)

Review form: Reviewer 1

Is the manuscript scientifically sound in its present form?

Yes

Are the interpretations and conclusions justified by the results?

Yes

Is the language acceptable?

Yes

Do you have any ethical concerns with this paper?

No

Have you any concerns about statistical analyses in this paper?

No

Recommendation?

Accept as is

Comments to the Author(s)

This manuscript reports studies of the reaction pathway of Mn(I)(CO)₃Br(bpCO₂) with H₂O₂. The studies reveal important insights into the factors (e.g., pH) that may influence the CO release reactivity. Overall, this manuscript is well written and provides understanding of the reactivity of a common family of CORMs. Differences in subcellular pH and H₂O₂ concentration may influence CO release. The insights into the relative reactivity of H₂O₂ and OOH⁻ with the Mn(I) CORM are an interesting contribution to the literature. I support publication with modification.

Review form: Reviewer 2**Is the manuscript scientifically sound in its present form?**

Yes

Are the interpretations and conclusions justified by the results?

No

Is the language acceptable?

Yes

Do you have any ethical concerns with this paper?

No

Have you any concerns about statistical analyses in this paper?

No

Recommendation?

Accept with minor revision (please list in comments)

Comments to the Author(s)

This manuscript by Barrett and co-workers describes a comprehensive mechanistic investigation into the H₂O₂-induced CO release from a manganese-containing carbonyl compound. There has been considerable recent interest in using such molecules as potential therapeutic agents and detailed insight into the processes that underpin their activity in an aqueous environment are extremely important in rationalising the behaviour and informing the design of future targets. The experimental work is carefully performed, mixing detailed kinetic studies and also the identification of the Mn-based species arising from CO release and I recommend publication when the following issues have been addressed.

1. The authors rightly highlight the species such as A are light-activated CO releasing molecules. It is not clear from the experimental section if the reaction kinetics and related mechanistic work were performed in the absence of light (e.g. using amberised glassware or using aluminium foil). In my view this is an important control to determine the role (or lack thereof) of light in the processes they are studying. For example, a kinetic experiment performed with and without the exclusion of light would enable this to be determined.
2. In my view the DFT calculations require further explanation. From the experimental section and supporting information it is not evident if the authors are quoting the electronic energy of the states they have calculated, or if any thermal corrections from a vibrational

spectrum calculation have been included. Indeed, they do not state if such a vibrational calculation has been performed. This is important to ensure firstly that the states are indeed minima on the potential energy surface but also, given the significant changes in molecularity during the calculated reactions, I would view determining the Gibbs energy changes as vital. I also note that the authors do not appear to have calculated any transition states for the processes shown in Figure 4 and 5. So I do agree with comments such as "Such calculations help in evaluating whether the respective intermediates are energetically accessible" but am less convinced by "Although DFT calculations suggest that reaction of A with either H₂O₂ or hydroperoxyl anion HOO⁻ to give E is a reasonable first step" in the absence of a calculated barrier to this process. It is certainly thermodynamically accessible as shown by the authors data, but no insight is provided into the kinetics which makes it difficult to determine if this really is a reasonable first step.

On a more trivial note, in a manuscript of this type I would expect the optimised coordinates for the calculated states to be presented in the Supporting Information so that others could repeat these calculations.

Decision letter (RSOS-211022.R0)

Dear Dr Barrett:

Title: Redox Mediated Carbon Monoxide Release from a Manganese Carbonyl – Implications for Physiological CO Delivery by CORMs.
Manuscript ID: RSOS-211022

Thank you for submitting the above manuscript to Royal Society Open Science. On behalf of the Editors and the Royal Society of Chemistry, I am pleased to inform you that your manuscript will be accepted for publication in Royal Society Open Science subject to minor revision in accordance with the referee suggestions. Please find the reviewers' comments at the end of this email.

The reviewers and handling editors have recommended publication, but also suggest some minor revisions to your manuscript. Therefore, I invite you to respond to the comments and revise your manuscript.

Because the schedule for publication is very tight, it is a condition of publication that you submit the revised version of your manuscript before 03-Sep-2021. Please note that the revision deadline will expire at 00.00am on this date. If you do not think you will be able to meet this date please let me know immediately.

When submitting your revised manuscript, you will be able to respond to the comments made by the referees and upload a file "Response to Referees" in "Section 6 - File Upload". You can use this to document any changes you make to the original manuscript. In order to expedite the

processing of the revised manuscript, please be as specific as possible in your response to the referees.

Kind regards,
Dr Ellis Wilde
Publishing Editor, Journals

On behalf of the Subject Editor Professor Anthony Stace and the Associate Editor Dr Annette Trunschke.

RSC Associate Editor
Comments to the Author:

I recommend that the manuscript be accepted, provided that the points noted by the reviewer(s) (in particular Reviewer 2) are addressed and a revised version is prepared on its basis.

RSC Subject Editor
Comments to the Author:
(There are no comments.)

Reviewer comments to Author:

Reviewer: 1

Comments to the Author(s)

This manuscript reports studies of the reaction pathway of $\text{Mn(I)(CO)}_3\text{Br(bpCO}_2\text{)}$ with H_2O_2 . The studies reveal important insights into the factors (e.g., pH) that may influence the CO release reactivity. Overall, this manuscript is well written and provides understanding of the reactivity of a common family of CORMs. Differences in subcellular pH and H_2O_2 concentration may influence CO release. The insights into the relative reactivity of H_2O_2 and OOH^- with the Mn(I) CORM are an interesting contribution to the literature. I support publication with modification.

Reviewer: 2

Comments to the Author(s)

This manuscript by Barrett and co-workers describes a comprehensive mechanistic investigation into the H_2O_2 -induced CO release from a manganese-containing carbonyl compound. There has been considerable recent interest in using such molecules as potential therapeutic agents and detailed insight into the processes that underpin their activity in an aqueous environment are extremely important in rationalising the behaviour and informing the design of future targets. The experimental work is carefully performed, mixing detailed kinetic studies and also the identification of the Mn-based species arising from CO release and I recommend publication when the following issues have been addressed.

1. The authors rightly highlight the species such as A are light-activated CO releasing molecules. It is not clear from the experimental section if the reaction kinetics and related mechanistic work were performed in the absence of light (e.g. using amberised glassware or using aluminium foil). In my view this is an important control to determine the role (or lack thereof) of light in the processes they are studying. For example, a kinetic experiment performed with and without the exclusion of light would enable this to be determined.
2. In my view the DFT calculations require further explanation. From the experimental section and supporting information it is not evident if the authors are quoting the electronic energy of the states they have calculated, or if any thermal corrections from a vibrational spectrum calculation have been included. Indeed, they do not state if such a vibrational calculation has been performed. This is important to ensure firstly that the states are indeed minima on the potential energy surface but also, given the significant changes in molecularity during the calculated reactions, I would view determining the Gibbs energy changes as vital. I also note that the authors do not appear to have calculated any transition states for the processes shown in Figure 4 and 5. So I do agree with comments such as "Such calculations help in evaluating whether the respective intermediates are energetically accessible" but am less convinced by "Although DFT calculations suggest that reaction of A with either H_2O_2 or hydroperoxyl anion HOO^- to give E is a reasonable first step" in the absence of a calculated barrier to this process. It is certainly thermodynamically accessible as shown by the authors data, but no insight is provided into the kinetics which makes it difficult to determine if this really is a reasonable first step. On a more trivial note, in a manuscript of this type I would expect the optimised coordinates for the calculated states to be presented in the Supporting Information so that others could repeat these calculations.

Author's Response to Decision Letter for (RSOS-211022.R0)

See Appendix A.

Decision letter (RSOS-211022.R1)

Dear Dr Barrett:

Title: Redox Mediated Carbon Monoxide Release from a Manganese Carbonyl – Implications for Physiological CO Delivery by CORMs.

Manuscript ID: RSOS-211022.R1

It is a pleasure to accept your manuscript in its current form for publication in Royal Society Open Science. The chemistry content of Royal Society Open Science is published in collaboration with the Royal Society of Chemistry.

Yours sincerely,
Dr Ellis Wilde
Publishing Editor, Journals

On behalf of the Subject Editor Professor Anthony Stace and the Associate Editor Dr Annette Trunschke.

RSC Associate Editor
Comments to the Author:
(There are no comments.)

Reviewer(s)' Comments to Author:

Appendix A

UNIVERSITY OF CALIFORNIA, SAN DIEGO & SANTA BARBARA

BERKELEY □ DAVIS □ IRVINE □ LOS ANGELES □ MERCED □ RIVERSIDE □ SAN DIEGO □ SAN FRANCISCO

SANTA BARBARA □ SANTA CRUZ

CELL: (310) 882-1787
email: jacob.alexander.barrett@gmail.com

Jacob A. Barrett
Former Chancellor's Postdoctoral Fellow (2019-2021)
466 Summer View Circle
Encinitas, CA 92024

September 17, 2021

Dr Ellis Wilde
Publishing Editor, Journals
Royal Society of Chemistry
Thomas Graham House
Science Park, Milton Road
Cambridge, CB4 0WF
Royal Society Open Science - Chemistry Editorial Office

Re: Manuscript ID: RSOS-211022

Dear Dr. Wilde:

We are returning the manuscript "Redox Mediated Carbon Monoxide Release from a Manganese Carbonyl—Implications for Physiological CO Delivery by CORMs." which has been revised in response to each of the comments by the reviewer. These revisions are described in the attached "Response to Reviewers". We thank the reviewers for their thoughtful comments.

Furthermore, in response to the five points raised in your letter.

- 1) A text file titled Mn_H2O2_Revision for the manuscript was uploaded.
- 2) Each figure as
- 3) A 100 word media summary was included in the revision
- 4) A dataset titled Raw Data for "Redox Mediated Carbon Monoxide Release from a Manganese Carbonyl—Implications for Physiological CO Delivery by CORMs" was submitted with DOI <https://doi.org/10.5061/dryad.mcvdnck1f> to the Dryad repository and uploaded with the revision.
- 5) All supplementary information was edited and proofread to ensure that it is legible and correct.

Sincerely,

Jacob Barrett
Peter C. Ford

Response to Reviewers:

Reviewer: 1

This manuscript reports studies of the reaction pathway of Mn(I)(CO)₃Br(bpCO₂) with H₂O₂. The studies reveal important insights into the factors (e.g., pH) that may influence the CO release reactivity. Overall, this manuscript is well written and provides understanding of the reactivity of a common family of CORMs. Differences in subcellular pH and H₂O₂ concentration may influence CO release. The insights into the relative reactivity of H₂O₂ and OOH⁻ with the Mn(I) CORM are an interesting contribution to the literature. I support publication with modification.

Response:

We thank the reviewer for their supporting comments.

Reviewer: 2

This manuscript by Barrett and co-workers describes a comprehensive mechanistic investigation into the H₂O₂-induced CO release from a manganese-containing carbonyl compound. There has been considerable recent interest in using such molecules as potential therapeutic agents and detailed insight into the processes that underpin their activity in an aqueous environment are extremely important in rationalising the behaviour and informing the design of future targets. The experimental work is carefully performed, mixing detailed kinetic studies and also the identification of the Mn-based species arising from CO release and I recommend publication when the following issues have been addressed.

1. The authors rightly highlight the species such as A are light-activated CO releasing molecules. It is not clear from the experimental section if the reaction kinetics and related mechanistic work were performed in the absence of light (e.g. using amberised glassware or using aluminium foil). In my view this is an important control to determine the role (or lack thereof) of light in the processes they are studying. For example, a kinetic experiment performed with and without the exclusion of light would enable this to be determined.

Response:

As the reviewer has noted, we indeed report that the substrate A is photoactive toward labilization of CO (see page 2 of the manuscript). We also had noted qualitatively that the reaction with H₂O₂ is accelerated by exposure to fluorescent room lights, although this reaction was not studied quantitatively. To avoid spurious photocatalysis, we aggressively minimized any exposure of reaction solutions to light. In this context we have added the following sentences to the Experimental section (changes noted in red):
bottom of p 2 top of p 3:

“Optical spectra and most kinetics were recorded on a Shimadzu UV-2401PC Spectrophotometer. **Actions were taken to limit any exposure of the cell to light. For example, most of the overhead room lights were off and the two remaining lights are equipped with amber filters. In addition, the spectrometer was covered with a black cloth to shield from the light from the monitor.** Liquid phase infrared spectra were recorded with a Nicolet iS50 FTIR spectrometer in solution cells with CaF₂ windows. Perpendicular-mode X-band EPR spectra were recorded for samples using in quartz tubes using a Bruker EMX EPR spectrometer equipped with an Oxford ESR 900 liquid helium cryostat. Aqueous samples were measured using a capillary insert to minimize the cross-sectional area of sample. Low temperature samples were frozen in a dry ice/ethanol bath prior to placement in the sample holder. **These EPR samples were shielded from light using a black cloth up until the point of measurement.**”

It was already noted in the experimental section that luminescence spectra were measured in a darkroom; however, we have added the following to the Experimental section paragraph on p 3 titled *Kinetics studies*:

“The reaction rates were determined from temporal absorbance changes using the Shimadzu UV-2401PC spectrophotometer with the samples protected from exposure to light as noted above.”

2. In my view the DFT calculations require further explanation. From the experimental section and supporting information it is not evident if the authors are quoting the electronic energy of the states they have calculated, or if any thermal corrections from a vibrational spectrum calculation have been included. Indeed, they do not state if such a vibrational calculation has been performed. This is important to ensure firstly that the states are indeed minima on the potential energy surface but also, given the significant changes in molecularity during the calculated reactions, I would view determining the Gibbs energy changes as vital.

Response: Thermal corrections were indeed included in the DFT calculations and we have revised the description of such calculations in the Experimental Section accordingly:

“Geometry and frequency calculations were performed with Gaussian 16 software package. The ground state structures were optimized at spin unrestricted B3LYP/DGDZVP level of theory without symmetry constraints, then the time-dependent DFT was used to obtain relative energy changes (ΔE , kcal/mol). All calculations in the ground and excited states were completed in the presence of a solvent (water) using Integral Equation Formalism of the Polarizable Continuum Model (IEFPCM). All energies are reported with thermal correction to 298 K.”

I also note that the authors do not appear to have calculated any transition states for the processes shown in Figure 4 and 5. So I do agree with comments such as “Such calculations help in evaluating whether the respective intermediates are energetically accessible” but am less convinced by “Although DFT calculations suggest that reaction of A with either H₂O₂ or hydroperoxyl anion HOO⁻ to give E is a reasonable first step” in the absence of a calculated barrier to this process. It is certainly thermodynamically accessible as shown by the authors data, but no insight is provided into the kinetics which makes it difficult to determine if this really is a reasonable first step.

Response: The reviewer is correct, transition states were not calculated. The goal of the DFT calculations were to determine whether the various intermediates proposed are energetically accessible as shown in the case of the formation of species E from species A. We agree that transition state calculations might provide some more insight; however, one would still have to assume a mechanism in order to make such calculations. The experiment subsequently described on pp 9 & 10, where the substitution of pyridine for the bromide of A strongly suppressed the reaction with H₂O₂, argues much more effectively that the lability of the bromide on A to give E is “a reasonable first step” in the reaction of A with H₂O₂.

On a more trivial note, in a manuscript of this type I would expect the optimised coordinates for the calculated states to be presented in the Supporting Information so that others could repeat these calculations.

Response: These coordinates have been added to the Supporting Information word document.